# Modulation of Pituitary Response by Dietary Lipids and Throughout a Temperature Fluctuation Challenge in Gilthead Sea Bream

**Sergio Sánchez-Nuño [1],\*** , **Sandra C. Silva [2]**, **Pedro M. Guerreiro [2]**, **Borja Ordóñez-Grande [1]**, **Ignasi Sanahuja [1]**, **Laura Fernández-Alacid [1]** and **Antoni Ibarz [1],\***

[1] Department of Cell Biology, Physiology and Immunology, Faculty of Biology, Universitat de Barcelona, Diagonal 643, 08028 Barcelona, Spain; b.ordonez@ub.edu (B.O.-G.); isanahuja@ub.edu (I.S.); fernandez_alacid@ub.edu (L.F.-A.)

[2] CCMAR—Centre for Marine Sciences, Universidade do Algarve, 8005-139 Faro, Portugal; scsilva@ualg.pt (S.C.S.); pmgg@ualg.pt (P.M.G.)

\* Correspondence: sancheznuno@ub.edu (S.S.-N.); tibarz@ub.edu (A.I.)

**Abstract:** Low temperatures provoke drastic reductions in gilthead sea bream (*Sparus aurata*) activity and nourishment, leading to growth arrest and a halt in production. However, scarce data exist concerning the implications of central core control during the cold season. The aim of this work was to study the effects of low temperature and recovery from such exposure on the pituitary activity of sea bream juveniles fed 18% or 14% dietary lipid. A controlled indoor trial was performed to simulate natural temperature fluctuation (22 °C to 14 °C to 22 °C). Meanwhile, we determined the regulatory role of the pituitary by analyzing the gene expression of some pituitary hormones and hormone receptors via qPCR, as well as plasma levels of thyroidal hormones. In response to higher dietary lipids, hormone pituitary expressions were up-regulated. Induced low temperatures and lower ingesta modulated pituitary function up-regulating GH and TSH and thyroid and glucocorticoid receptors. All these findings demonstrate the capacity of the pituitary to recognize both external conditions and to modulate its response accordingly. However, growth, peripheral tissues and metabolism were not linked or connected to pituitary function at low temperatures, which opens an interesting field of study to interpret the hypothalamus–pituitary–target axis during temperature fluctuations in fish.

**Keywords:** low temperatures; pituitary hormones; *Sparus aurata*; temperature recovery; thyroxine; triiodothyronine

## 1. Introduction

Fish are ectotherms and thus their physiology is directly influenced by fluctuations in water temperature, which can be dramatic and fatal according to the range of their thermotolerance. Drops and sustained low water temperatures may lead to winter mortalities; a phenomenon already reported for a large number of fish species (reviewed in [1]). Although extreme cold shock is, in some cases, the direct cause of mortality, growing evidence indicates that thermal stress and starvation are the major underlying physiological conditions that cause mortality in fish during winter [2].

In aquaculture, despite the significant bottleneck created by cold-suppressed appetite and growth, relatively few studies of these phenomena exist. In the Mediterranean Sea, wild gilthead sea bream (*Sparus aurata*) normally experience seasonal water temperature variations ranging from 11 °C in winter to 26 °C in summer, and the fish migrate to greater depths and warmer waters when surface temperatures start to decline [3]. However, in intensive culture, fish are unable to migrate and the

temperature decrease during winter can become critical (reviewed in [4]). At low temperatures, fish activity and growth are minimal [5–8], and a drastic reduction in nourishment is observed, leading to growth arrest and a halt in production [6].

Although compensatory responses have been demonstrated in many fish, there is no significant thermal compensation for cold conditions in gilthead sea bream at temperatures below 13–14 °C (reviewed in [4]). The first sign of cold stress in fish is reduced food intake [9] and, in sea bream, the reluctance to feed does not depend on the rate of temperature drop and normal feeding is not resumed during extended periods at constant low temperatures [10,11]. As a consequence, physiology is impacted and both global metabolic rate and hepatic enzyme activities are greatly depressed at low temperatures [5,10,12,13]. Osmoregulatory capacity is also impaired, showing seasonal dependence under culture conditions, with lower values of osmolality and main ion concentration in winter than in summer [14,15]. Nonetheless we have seen that dietary supplementation during temperature recovery promotes better liver condition, and maintains osmoregulatory function and bone formation [16].

The overall control of external inputs, internal conditions and body responses is mediated by the hypothalamus–pituitary axis: a central core which regulates peripheral behaviour. The pituitary gland is an intermediary organ for physiological signal exchanges between the hypothalamus and peripheral organs, and it acts as a key regulator during development, stress, and other physiological processes, either by producing major circulatory hormones such as growth hormone, prolactin or somatolactin, or by regulating the circulating levels of cortisol and thyroid hormones. However, few studies refer to the winter hormonal inhibition of feeding and growth in gilthead sea bream, and, to the best of our knowledge, no studies have addressed the action of the pituitary gland during winter growth arrest and recovery in gilthead sea bream. An increased release of somatotrophic hormones has been reported, always preceding warm growth spurts, whereas somatolactin circulating levels rose in late autumn, concurrently with low plasma cortisol levels [17]. In gilthead sea bream, as in other fish species, the circulatory levels of the two thyroid hormones, triiodothyronine ($T_3$) and tetraiodothyronine or thyroxine ($T_4$), are reduced during induced food restriction, whereas only $T_3$ recovers after a week of refeeding [18]. However, no such data exist on the effects of low temperature for this species. Moreover, low temperatures and induced fasting are also reported to be stressor factors, whether achieved gradually or acutely [4]. The stress response is also driven by the pituitary via expression of proopiomelanocorticotropin (POMC) derived peptides. Whereas extensive literature links the stress axis to fish confinement conditions (fish density or crowding, handling, etc.), only one study under laboratory conditions has reported the interaction between low temperatures and the pituitary–inter-renal axis in sea bream; showing a rapid, but transient, release of cortisol and adrenocorticotropin after a drop in water temperature from 18 °C to 9 °C [19]. As explained below, no data previously existed on the direct pituitary response to environmental challenges such as low temperatures and recovery under culture conditions, one of the main concerns for gilthead sea bream production [4].

The advantages of modified fish diet formulation at low temperatures is still a relevant and controversial concern in sea bream production. High quality lipids (fish oil rich in polyunsaturated fatty acids) promote fish growth, but diets with high lipid content in winter or the pre-winter season appear to increase the incidence of pathologies associated with low temperatures [4]. We recently observed that reducing dietary lipids, and consequently dietary energy, did not affect gilthead sea bream performance during controlled thermal fluctuation, but did alter both intermediary metabolism and redox status in liver [8,16,20]. Dietary lipids, as other nutrients, influence the endocrine factors that regulate feeding and growth (recently reviewed in [21]) and, despite the lack of growth, there may be changes in hormonal gene expression concerning dietary formulation in several fish species. Some authors suggested that the synthesis and secretion of ACTH-induced cortisol in interrenal cells of sea bream is conditioned by essential lipids [22], and showed in pejerrey juveniles (*Odontesthes bonariensis*) that there is a differential response of the growth hormone (GH)-IGF system to cope with changes in dietary lipid composition, despite the lack of a clear somatic growth response [23].

Most of the studies mentioned above dealt with circulatory levels of target hormones, but not the regulatory role of the pituitary itself, and no studies have reported pituitary functionality during low temperatures and recovery challenges, or their implications for winter growth arrest in gilthead sea bream cultures. Therefore, the present work studies, for the first time, gilthead sea bream pituitary modulation in response to an induced temperature fluctuation, divided in three periods: pre-cold (30 days at 22 °C), cold (50 days at 14 °C), and recovery (35 days at 22 °C), and under two dietary regimens with total lipid content of 18% and 14%. To determine pituitary function, we measured gene expression for hormone and receptors by quantitative real-time PCR (qPCR), as indicators of central response to dietary lipid change and temperature fluctuation. The selected hormones were: thyroid-stimulating hormone (TSH); GH; and two forms of proopiomelanocorticotropin (POMC), the so-called sbPOMCα1 or POMCA and sbPOMCα2 or POMCB (reported in gilthead sea bream in [24]). The pituitary receptors selected were the glucocorticoid receptor (GR), and thyroid receptors (TRα and TRβ). Moreover, levels in plasma thyroid hormones $T_3$ and $T_4$ were also measured and taken in account together with known data on growth [8] and plasma metabolites [16] via principal components analyses (PCA).

## 2. Results

### 2.1. Effects of Dietary Lipid on Pituitary Expression

Expression of pituitary hormones *tsh*, *gh*, *pomca* and *pomcb* and the hormone receptors *trα*, *trβ*, and *gr* have been used as indicators of central response to dietary lipid change and temperature variations. Daily rhythm interactions with these pituitary markers were avoided by always obtaining pituitary samples at same time in the morning, before the first daily ingesta. Thus, pituitary expression levels could be compared between all the sampling moments. Relative expressions of hormones and hormone receptors were normalised by D14 pre-cold values in all cases. Figure 1 shows relative gene expression of pituitary hormones *gh*, *tsh*, *pomca* and *pomcb* and Figure 2 shows relative gene expression of pituitary receptors *trα*, *trβ*, and *gr* in response to dietary lipid adaptation and with the concomitant energy differences. After 30 days at 22 °C, the pituitary of the animals fed with D18 had up-regulated the expression of hormones in response to the greater amount of energy in the diet. GH and TSH, both hormones directly related to metabolism, increased it expression by 3- and 6-fold (Figure 1A,B), respectively. The melanocortin system, as measured by POMC peptides (*pomca* and *pomcb*), was also significantly increased—4- and 6-fold, respectively (Figure 1C,D). In addition to hormonal expression, the pituitary has multiple receptors that allow response to the different inputs received. With regard to the thyroid hormone receptors, their expression differed: *trα* was up-regulated 3-fold in D18 and *trβ* did not change its relative expression in response to dietary lipids (Figure 2A,B). Data on Figure 2 showed that the *gr* was over-expressed as a result of diet D18 (Figure 2C), in the same way as hormone expression.

### 2.2. Effects of Thermal Fluctuation on Pituitary Expression

Pituitary activity during the thermal fluctuation is shown in Figure 3 (hormones) and Figure 4 (hormone receptors). Despite the expected metabolic depression due to low temperatures (14 °C), the pituitary was capable of modulating hormone expression, up-regulating *gh* and *tsh* at the end of 50 days at 14 °C, showing a clear cold acclimation response, irrespective of diet (Figure 3A,B). In contrast, neither *pomc* peptide expression level was modified (Figure 3C,D). With regard to the hormone receptors studied, maintained cold influenced them differently, depending on diet, with all the receptors studied significantly increased for D14 (Figure 4). For the high lipid diet, only *trβ* was up-regulated significantly. In summary, despite the lower intake observed during the cold period, more lipids in the diet caused over-expression of the pituitary hormones analysed, whereas dietary lipid reduction provoked higher expression of pituitary hormone receptors.

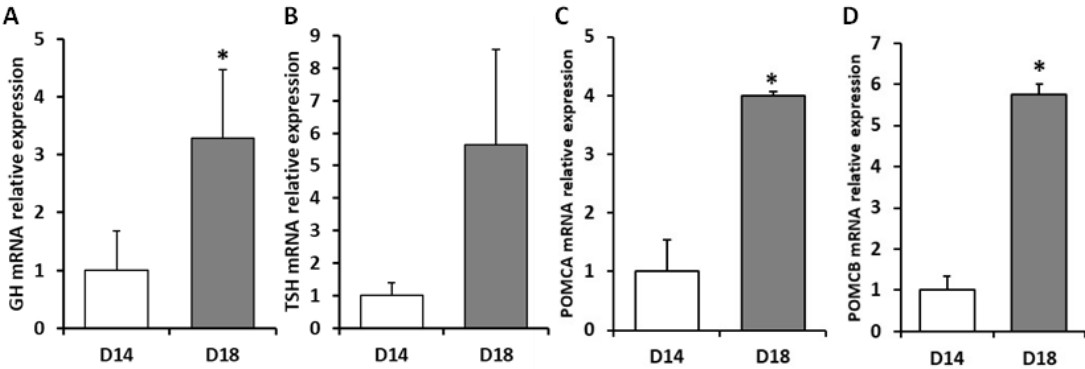

**Figure 1.** Relative gene expression of pituitary hormones of gilthead sea bream juveniles fed two dietary regimes for 35 days at 22 °C. Data are shown as arbitrary units referred to relative expression for D14 as Mean ± Standard error of mean. Hormones analysed were: (**A**) Growth Hormone, GH; (**B**) Thyroid Stimulating Hormone, TSH; (**C**,**D**) Proopiomelanocorticotropin peptides A and B, POMCA and POMCB. * indicate significant differences between D14 and D18 (Student's *t*-test).

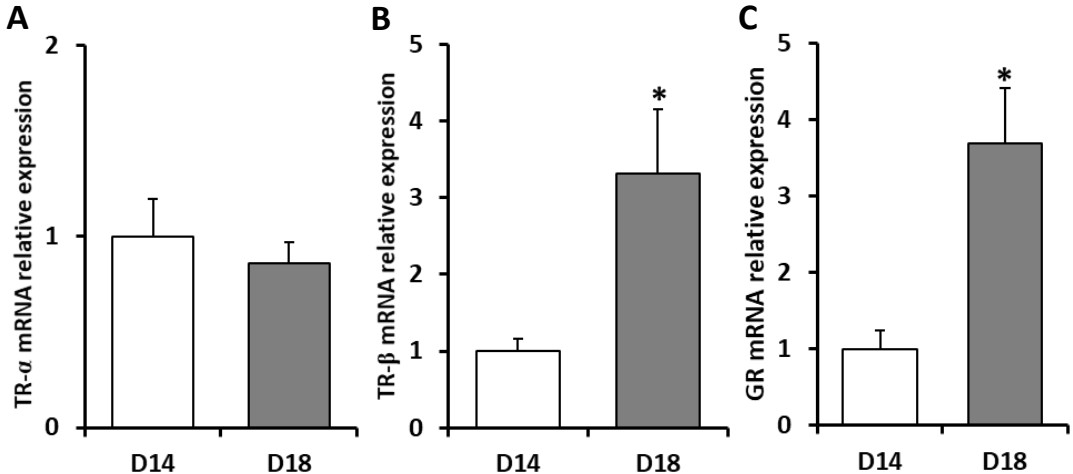

**Figure 2.** Relative gene expression of pituitary receptors of gilthead sea bream juveniles fed two dietary regimes for 35 days at 22 °C. Data are shown as arbitrary units referred to relative expression for D14 as Mean ± Standard error of mean. Receptors analysed were: (**A**,**B**) Thyroid hormone Receptors alpha and beta, TR-alpha and TR-beta; (**C**) Glucocorticoid Receptor, GR. * indicate significant differences between D14 and D18 (Student's *t*-test).

To better understand the recovery process from low temperatures and low ingesta at the pituitary level, two samplings were performed during the recovery period: at day 7 after the start of the recovery process (early recovery) and after 35 days at 22 °C (late recovery). Whereas the over-expressed *tsh* values during cold had reverted to pre-cold values already during early recovery, *gh* expression was still higher than the pre-cold values during early recovery and levels recovered later—again, irrespective of diet. Expression of genes for the POMC peptides responded to temperature increase differently, according to diet. For D14, both expressions increased more than 2-fold during early recovery, whereas, for D18, *pomca* was down-regulated and *pomcb* was not significantly increased. These data demonstrate a diet effect on the pituitary response to cope with thermal restoration during early recovery. The hormone expression changes were all reverted to pre-cold values for both diets during late recovery. Thyroid hormone receptors and *gr* were reverted later for D14, with the expression levels being higher during early recovery—even *trα* increased. Curiously, for D18, *trα* expression levels did not revert at the end of the trial, and transcripts were over 2-fold higher than in the pre-cold condition.

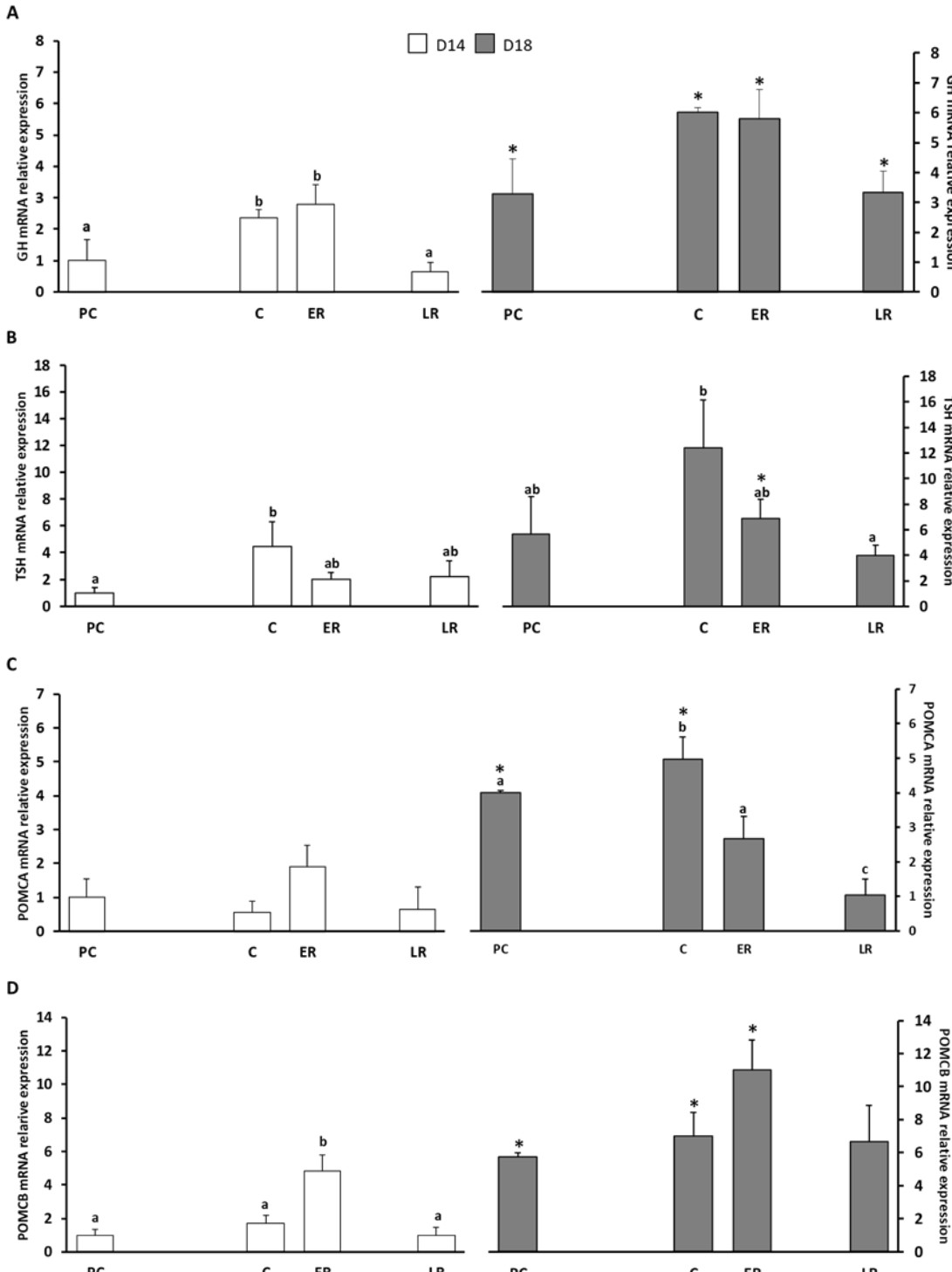

**Figure 3.** Relative gene expression of pituitary hormones of gilthead sea bream juveniles fed two dietary regimes during temperature fluctuations. White bars correspond to animals fed with D14, while grey correspond to animals fed with D18 diet. Data are shown as arbitrary units referred to relative expression for D14 as Mean ± Standard error of mean. PC: Pre-Cold; C: Cold; ER: Early Recovery; LR: Late Recovery. Hormones analysed were (**A**) Growth Hormone, GH; (**B**) Thyroid Stimulating Hormone, TSH; (**C**,**D**) Proopiomelanocorticotropin peptides A and B, POMCA and POMCB. Lower-case letters indicate significant differences between periods (one-way ANOVA); * indicate significant differences between D14 and D18 (Student's *t*-test).

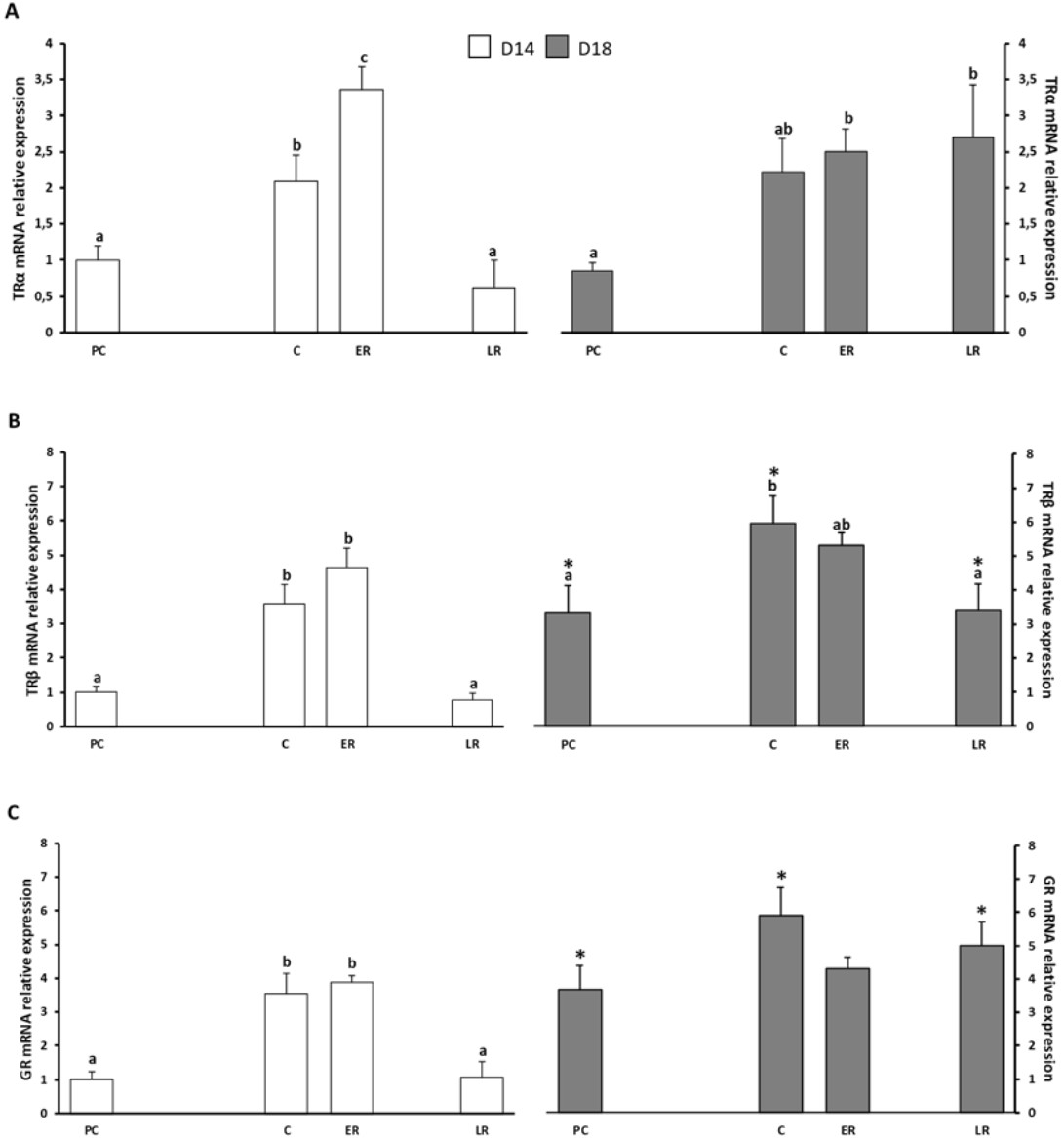

**Figure 4.** Relative gene expression of pituitary receptors of gilthead sea bream juveniles fed two dietary regimes during temperature fluctuations. White bars correspond to animals fed with D14 while grey to animals fed with D18 diet. Data are shown as arbitrary units referred to relative expression for D14 as Mean ± Standard error of mean. PC: Pre-Cold; C: Cold; ER: Early Recovery; LR: Late Recovery. Receptors analysed were: (**A**,**B**) Thyroid hormone Receptors alpha and beta, TR-alpha and TR-beta; (**C**) Glucocorticoid Receptor, GR. Lower-case letters indicate significant differences between periods (one-way ANOVA); * indicate significant differences between D14 and D18 (Student's *t*-test).

*2.3. Effects of Dietary Lipids and Thermal Fluctuation on Plasma Thyroid Hormones*

Circulating thyroidal hormones were measured by respective ELISA tests and values corresponded to levels after overnight fasting and before the first daily ingesta. Levels of the active form $T_3$ and the prohormone $T_4$, as well as the $T_4/T_3$ ratio, as an indicator of peripheral transformation of $T_4$ to $T_3$, are shown in Table 1. Data for fish maintained at 22 °C show circulating levels around 19–21 ng/mL for $T_4$ and 4–5 ng/mL for $T_3$, resulting in a $T_4/T_3$ ratio between 4 and 4.5, irrespective of dietary lipid amount. Low temperatures provoked a significant rise in the $T_4/T_3$ ratio, due to lower levels of circulating $T_3$ after overnight fasting, suggesting reduced peripheral transformation at low temperatures with respect to pre-cold values. Again, this effect was not diet-dependent. Temperature and ingesta recovery

partially restored levels of circulating hormones in a week (early recovery), which was more evident in $T_4$ and provoked an over-release at the end of the recovery period (late recovery). This recovery period significantly affected the $T_4/T_3$ ratio, restoring its pre-cold values.

**Table 1.** Thyroid hormones levels of juvenile gilthead sea bream throughout temperature fluctuation feeding different dietary lipid levels.

| Thyroid Hormones | Pre-Cold (Day 30) | Cold (Day 80) | Early Recovery (Day 87) | Late Recovery (Day 115) |
|---|---|---|---|---|
| T3 (ng/mL) | | | | |
| D14 | 4.4 ± 0.9 [a] | 2.2 ± 0.4 [a] | 3.7 ± 0.6 [a] | 6.9 ± 1.1 [b] |
| D18 | 5.1 ± 0.8 [ab] | 2.5 ± 0.4 [a] | 4.1 ± 0.7 [ab] | 7.2 ± 2.0 [b] |
| T4 (ng/mL) | | | | |
| D14 | 19.2 ± 0.5 [a] | 19.9 ± 0.2 [a] | 21.1 ± 0.4 [b] | 22.5 ± 0.4 [c] |
| D18 | 20.8 ± 0.5 | 20.9 * ± 0.2 | 20.7 ± 0.5 | 20.5 * ± 0.3 |
| Ratio T4/T3 | | | | |
| D14 | 4.5 ± 0.7 [ab] | 8.9 ± 0.3 [c] | 6.4 ± 0.9 [b] | 3.3 ± 0.4 [a] |
| D18 | 4.0 ± 0.1 [a] | 7.7 ± 0.8 [b] | 6.0 ± 1.2 [ab] | 3.9 ± 1.3 [a] |

Data are expressed as Mean ± S.E.M. Lower-case letters indicate significant differences between thermal periods within each diet ($p < 0.05$, One-Way ANOVA). Significant differences between diets at each thermal period are indicated by * (Student's *t*-test).

## 3. Discussion

### 3.1. Dietary Modulation of Pituitary Response at Warm Temperature

Feed is the greatest single cost in fish farming and feed costs become even more important. Beyond feed costs, fish diet should also be an active way to cope with the effects of seasonal challenges on culture. Recent studies demonstrate the negative interaction between the excessive dietary energy levels, the culture density on growth performance, and the metabolic and oxidative status [25]. In the case of gilthead sea bream, reducing dietary lipid before the cold season has been proposed to cope with liver affectations [4,26–28]. We previously demonstrated that reducing dietary lipid from 18% to 14% resulted in a similar body weight, condition factor, SGR (0.86 ± 0.05 for D14 and 0.97 ± 0.05 for D18) and FCR (1.32 ± 0.02 for D14 and 1.33 ± 0.06 for D18) (data from [16]), with an improved redox status [8,16], suggesting a more efficient metabolism when sea bream have been fed a low lipid diet. The study of pituitary function during the pre-cold period showed that dietary lipid intake is capable of modifying pituitary expression of *gh*, *tsh* and both *pomc* hormones, together with that of the receptors *trβ* and *gr*. Thus, the pituitary responded to dietary inputs, showing higher expression for the diet with higher lipids and energy, always measured before the programmed morning feed intake.

There is little information on the regulation of the gene expression of our selected pituitary markers in relation to dietary response and temperature fluctuation, and, as far as we know, no data exist for the response of this gland to the two factors combined. In a previous study, we determined an increased cellular activity of the pituitary in response to amino acid supplementation, as measured via proteome expression profiles [29]. However, relating peripheral effects with pituitary expression would contribute to better understanding the present results.

In fish, it well established that the control of pituitary secretion is mutifactorial, with several stimulatory and inhibitory neurohormones acting directly on somatotrophs, thyrotrophs or corticotrophs. This modulation is also dependent on several external stimuli, including feeding regimes or nutritional value [30]. Gene expression of growth hormone, a major pituitary maker for growth and satiation, was reduced by feeding energetic restriction in cultured *Labeo rohita* [31], and in the present study, *gh* expression was up-regulated when the fish were fed diet D18, which is in agreement with the increased circulating levels of *gh* observed in gilthead fingerlings fed with 17% lipid with respect to 9% [32]. However, in the present case and contrary to those studies, the change in *gh* expression did not result in significant changes in the analysed growth parameters during this

30-day period [8]. Growth hormone actions are mediated by the Insulin growth factor 1 (IGF-1), which was not evaluated in relation to diet. Interestingly, however, the differences in expression between D14 and D18 are coincidental with changes in the activity of liver glucose-6-phosphate dehydrogenase (G6PDH) [16]. Although we have not analysed circulating GH, this hormone has been shown to be a potent regulator of the enzyme gene expression and activity in liver in silver sea bream (*Sparus sarba*), independently of IGF-1 [33,34].

Thyroid stimulating hormone is critical in regulating post-prandial responses in fish, by the release of thyroid hormones. Despite marked differences between *tsh* expression for D18 and D14, circulating levels of $T_3$ and $T_4$ did not differ, which was possibly related to the pre-ingesta condition, whereby the eating signal was not yet present, as reported in mammals [35,36]. Moreover, in red drum (*Sciaenops ocellatus*), it was suggested that TSH follows a dynamically regulated 24 h rhythm [37] and, in sea bream, the presence of plasma $T_3$ and $T_4$ peaks were reported, influenced by the feeding moment [38].

In sea bream, a differential distribution of the expression of *pomc* transcripts in pituitary (*pomc-α1* or *pomca* and *pomc-α2* or *pomcb*) was reported [24]. In our study, both transcripts of POMC are up-regulated in response to higher lipid input. It is well established that hypothalamic POMC neurons are responsive to the nutritional status and act as strong anorexigenic regulators [39,40] but the role of pituitary POMC in food metabolism is not clear. Assuming a similar involvement, a higher lipid diet should lead to a longer transient decrease in appetite, consistent with higher *pomc* levels in the pre ingesta period. Because the responses derived from pituitary *pomcs* expression depend on the production of many biological peptides such as melanocyte-stimulating hormones (MSHs), corticotrophin (ACTH) and β-endorphin [41], to distinguish the interesting role of each POMC and the role of their products in response to dietary modifications would need further research.

Thus, although the changes in gene transcription at the pituitary level in relation to feed energy are important, it is not clear if they may result in altered hormone secretion and overall impact. To better elucidate the peripheral effects (growth and plasma markers) of dietary lipids proposed in the current trial, we performed a PCA analyses (Figure 5) of body performance data, such as body weight, body length, condition factor and hepatosomatic index [8] together with relevant plasma metabolites markers such as glucose, lactate, and lipid fractions (NEFA and TAG) [16]. Similarly to thyroid hormone levels, we observed that both body parameters and plasma metabolites such as glucose, lactate or proteins did not show differences after overnight fasting for fish fed 18% or 14% dietary lipids [8,16], and resulting PCA did not discriminate between D14 or D18, differently to above-mentioned greater differences in pituitary expression.

The amounts of specific hormone receptors are crucial for hormone release regulation of central nervous control via feedback loops [42,43]. In response to higher dietary lipid, *trβ* and *gr* expressions increased, whereas *trα* was the only pituitary marker to show no differences between D18 and D14 during pre-cold. In fish, TRs are related with multiple physiological processes and present isoforms [44], whereas in mammals *trβ* transcripts show higher levels than *trα* in anterior pituitary [45]. This finding would explain the higher sensitivity of *trβ* modulation by diet compared with *trα* in the current study, in agreement with the idea that *trβ* gene products may mediate thyroid hormone feedback regulation of *tsh* [45]. Glucocorticoid receptor *gr* expression in sea bream has been suggested to be tissue specific and correlated to plasma cortisol levels [46]. This could not be the case in our pre-cold conditions, where a change in total dietary lipid of 4% did not prove to be a stressor condition [8,16]. As reported for pituitary modulation by amino acid supplementation [29], dietary lipids were also recognized as conditioning pituitary function by the expression of both the hormones and receptors studied. However, whereas the external input from diet affects the pituitary, whole animal responses seem to compensate for the higher energy by adapting metabolism, which results in no differences in growth [16]. Further studies are necessary to better determine the effects of pituitary modulation and the lack of growth differences between dietary regimes.

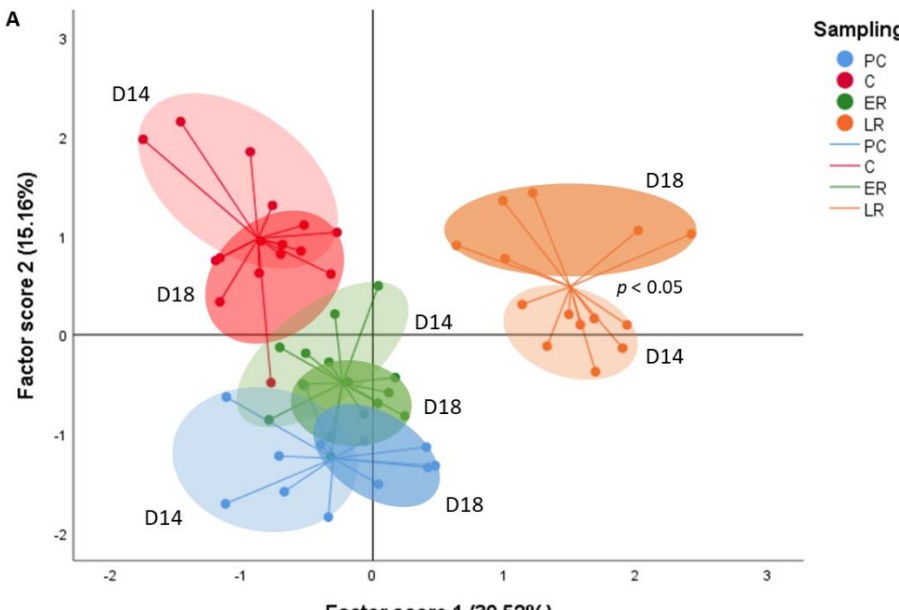

**Figure 5.** Principal Component Analysis (PCA) plot of body parameters, plasma metabolites and plasma thyroid hormones throughout thermal fluctuation and dietary lipid challenges. Data from body performance (body weight, body length, visceral fat, condition factor and hepatosomatic index) correspond to data reported in [8]. Data from plasma markers (glucose, lactate, triacyclglycerides, non-esterified fatty acids) correspond to data reported in [16]. (**A**) Coloured clouds indicate each thermal period (Pre-Cold, PC; Cold, C; Early Recovery, ER; Late Recovery, LR). Compact coloured clouds correspond to D18 individual fish and blur coloured clouds correspond to D14 individual fish. Factors 1 and 2 represent the first and second principal components. Axis parentheses indicate the factors explained variance. (**B**) Indicate the factor contribution of each variable and (**C**) indicate the component variance accumulation to a maximum of 90%.

*3.2. Cold Effects*

For gilthead sea bream, cold season growth arrest and cold-induced metabolic depression are unavoidable under rearing conditions and affect the whole animal, globally (reviewed in [4]), and specifically liver intermediary metabolism [16] and redox metabolism [8,20]. The growth arrest induced by the low temperature of 14 °C was evidenced by the drastic reduction in the SGRs (around 4-fold) and the doubling of the FCRs (data from [16]). However, the central core mechanisms underlying the overall response are still unknown. The present study is the first attempt to elucidate whether the pituitary in gilthead sea bream is capable of detecting water temperature decrease. Despite the drastic fall in peripheral tissue responses to sustained cold exposure, pituitary markers revealed a

generalized up-regulation of their gene expression, irrespective of diet. As cold and reduced intake occur at the same time, it is difficult to know how pituitary, or, higher up, the hypothalamus, interprets the concomitant signals. In silver sea bream, higher amounts of pituitary *gh* transcript and content were reported in chronic acclimation at 12 °C, in comparison to those maintained at 25 °C [47]. In rabbitfish (*Siganus guttatus*), higher *gh* mRNA levels were reported in the pituitary after 15 days of starvation [48]. In agreement with these results, gilthead sea bream at 14 °C for 50 days doubled their pituitary *gh* expression, for both dietary regimes. With regard to *tsh* expression, our results indicate more than a 2-fold up-regulation, irrespective of diet, at the end of the cold period. Although there are no data on pituitary *tsh* expression in sea bream, it was reported that 45 days of cold exposure at 6 °C decreased plasma TSH in common carp (*Cyprinus carpio*), with plasma hormonal changes, due to cold stress already evident at day 15 [49]. Beyond low temperatures, starvation seems to provoke a lack of thyroid tissue sensitivity to TSH, eliminating daily plasma TH patterns and monodeiodinase activity [18,50,51]. We observed that circulating $T_3$ was diminished at low temperatures, thereby increasing the $T_4/T_3$ ratio. This ratio has been proposed as a marker of peripheral deiodinase activity [51], transforming $T_4$ into the active hormone $T_3$ [52]. Thus, according to the observed peripheral enzymatic depression, after 50 days of cold exposure [8,16,20], low temperatures of 14 °C should unbalance the $T_4/T_3$ ratio by also diminishing peripheral tissue deiodinase activity, again irrespective to dietary lipids. Although further studies are needed, the reduction in circulating $T_3$ could explain the increment of both thyroidal receptors in pituitary at low temperatures, indicating that the pituitary is capable to integrate internal inputs (feedback) as well as external inputs. Considering together the effects on body parameters, plasma markers and plasma thyroid hormones, the PCA plot clearly discriminated "cold fishes" from pre-cold, mainly due to thyroid hormone ratio NEFA and TAG levels, as was expected. However, no differences could be drawn from D14 and D18 fishes. These data increased the controversy between central core modulation of pituitary enhanced expression and the lack of adaptation of peripheral tissues.

Although temperature fluctuations are natural for fish, abrupt changes may become an effective stressor, inducing classic stress responses mediated by pituitary expression of *pomca* and *pomcb*. [53] reported a cortisol peak after acute or chronic stress in juvenile chinook salmon (*Oncorhynchus tshawytscha*) acclimated to three different temperatures. It was demonstrated that acute temperature changes amplified the endocrine response to handling and confinement [54], while, studying the acclimation of Atlantic cod (*Gadus morhua*) to cold water, it was stated that cold water exposure caused a marked increase in secondary responses such as plasma cortisol and glucose concentrations [55]. However, after 50 days at 14 °C, sea bream did not show over-expression of *pomc* peptides, or any secondary effects of the HPI axis. These controversial results could be in accordance with the scarce data on cold stress in sea bream, where a transitory and unsustained release of cortisol and ACTH were reported after a drop in water temperature from 18 °C to 9 °C [19]. In contrast, pituitary GR increased at low temperatures, which would suppose the presence of higher amounts of circulating cortisol [46]. Further, it has been suggested that POMC, through the actions of α-MSH and cortisol, may have an inhibitory effect on food intake (reviewed in [40]). In our case, it is likely that the overall loss in appetite may be partially triggered by this axis, which, as indicated above, is activated by the cold shock, but since our samples were not collected immediately after the drop, but after 50 days at low temperature, it is not possible to evaluate this involvement. Further studies on the chronic effects of low temperatures should relate the stress axis with the scarce signals of stress response when the fish metabolism is inhibited by cold conditions.

### 3.3. Temperature Recovery

The capacity of fish to recover from cold periods is another focus point for fish farmers, and sea bream producers in particular [4]. During the recovery period, growth performance was not completely restored, with the SGRs from the recovery period being lower (between 0.70%−0.74% of body weight gain per day) than those from the PC period, and the FCR values still being higher (between 1.72−1.76

g of feed per gram of body weight) (data from [16]). Although few studies have addressed the recovery dynamics of growth and fish physiology, none have considered pituitary functionality. In this work, we analysed, for the first time in fish, the effect of temperature recovery on pituitary response and plasma $T_3$ and $T_4$. To better understand the recovery process, two samplings were carried out to evaluate the pituitary response and circulating thyroid hormones. As occurs during exposure to low temperatures, the pituitary detected the external input of warming and ingesta restoration. It is striking that, unlike in previous periods, the gradual reestablishment of the temperature to 22 °C modulated each of the pituitary markers analysed differently: *tsh* expression recovered pre-cold expression levels in 7 days; *gh*, both *trα* and *trβ*, and *gr* expression had recovered after 35 days of recovery; and *pomc* showed different responses for D14 and D18. The recovery of *tsh* expression was achieved in one week at 22 °C, suggesting the importance of recovering normal TSH release early, although the consequences of this mechanism have not yet been described in fish (reviewed in [52]). The *gh* down-regulation was delayed with respect to *tsh*. The increase in temperatures also increases fish food intake [8,16] and this restoration is strongly related with both *tsh* and *gh* up-regulation, due to higher ingesta and energy intake. Plasma levels of $T_3$ showed an overcompensation at the end of the recovery period, showing the relevance of thyroidal hormones on the recovery process, and this could be a signal of the presence of compensatory growth [52]. Accordingly, the PCA analyses revealed the differences existing between early recovery, where animals approached pre-cold values, and late recovery, where animals clearly grouped away from pre-cold. Only in the case of later recovery did PCA significantly separate D14 and D18 animals. Meanwhile, the expression of the thyroid receptors also suggested an inherent diet effect. In D14, both *trα* and *trβ* were still over-expressed in early recovery, only returning to initial values at the end of the recovery period.

With regard to the expression of POMCs, the recovery process elucidates different patterns on *pomca* and *pomcb* down-regulation, and they were diet-dependent. Due to the lack of knowledge on the regulatory mechanisms of *pomc* transcript expression, it is difficult to establish their behaviour. Warming caused an increment of *pomc* transcripts in D14, indicating a stressor effect of this diet, in agreement with previous data reflecting the lower capacity of animals fed lower lipids to cope with temperature restoration [8,16]. Moreover, pituitary expression of *gr* in D14 was delayed longer than in D18, supporting a feedback signal of a stressor condition for that diet.

## 4. Materials and Methods

To study the effects of the cold season on the pituitary capacity of gilthead sea bream to cope with environmental changes, we performed a controlled indoor trial that simulated natural thermal fluctuations, composed of three periods: the start of winter (pre-cold period), winter (cold period) and leaving winter behind (recovery period). Two isoproteic diets, one with a reduced dietary lipid content (from 18% to 14%), were introduced to evaluate the effects of dietary energy throughout the temperature fluctuations. This study focused on the regulatory role of the pituitary by analysing the gene expression of some pituitary markers by qPCR and the plasma levels of thyroidal hormones by ELISA-Kit. To avoid interference on these expressions and circulating hormones from daily rhythms, all samples were collected after overnight fasting, before the morning feed intake.

### 4.1. Animal Condition and Sampling

Gilthead sea bream, 145 g average body weight, from a local fish farm were acclimated indoors at the Centre d'Aquicultura (CA-IRTA, Sant Carles de la Ràpita, Tarragona, Spain) at 22 °C for two weeks using standard commercial fish feed (Skretting_ARC). Following this period, they were randomly distributed into two experimental groups, each in triplicate (30 fish per tank), and held in a water recirculating system: IRTAmar™. Six 500-L fiberglass tanks were used, monitoring the solid and biological filters, water temperature, and oxygen concentration, while maintaining, throughout the experimental period, nitrite, nitrate and ammonia concentrations at initial values. Two isoproteic diets formulated by Skretting ARC, one with a lower lipid content than the other (reduced from 18% to 14%;

diets D18 and D14, respectively) were used, corresponding to a "crude energy" of 21.0 and 20.2 MJ/kg dry matter, respectively. The fish were fed to satiation twice a day, 7 days per week for 115 days—the total experimental period. Feed was automatically delivered for each 30 min feed session at 0800 h and 1600 h. Satiation was ensured throughout experimental period by calculating estimated daily feed intake and allowing a ration 20% above this value. Feed delivery was recorded daily and uneaten feed was collected daily by the system and then manually dried and weighed to calculate real feed intake (more details in [16]).

The experimental period (115 days) consisted of four periods: pre-cold (PC), during which the fish were maintained at 22 °C for 30 days; cold (C), during which the temperature was cooled to 14 °C over five days and maintained at this temperature for an additional 45 days (giving a total of 50 days); early recovery (ER), during which the temperature was restored to 22 °C over five days and measurements were taken seven days after the start of the temperature recovery period; and late recovery (LR), for which measurements were taken 35 days after the start of temperature recovery. At the end of each temperature period, fish were fasted overnight before sampling and nine fish (3 per replicated tank and diet) were captured at random and anaesthetised with 2-phenoxyethanol (100 ppm) diluted in seawater. Body weight and length were measured, blood samples were taken from the caudal vessels using EDTA-Li as the anticoagulant, and the fish were killed by severing the spinal cord. Plasma was obtained by centrifuging the blood at 13,000 $g$ at 4 °C for 5 min and samples were kept at -80 °C until analysis. The study complied with the guidelines of the European Union (86/609/EU), the Spanish Government (RD 1201/2005) and the University of Barcelona (Spain) regarding the use of laboratory animals.

## 4.2. Plasma Thyroid Hormones

The thyroid hormones $T_3$ and $T_4$ were measured in plasma by ELISA following the guidelines of the commercial kits (Cusabio Biotech Co., Houston, TX, USA). These assays employ the competitive inhibition enzyme immunoassay technique using microtiter plates pre-coated with specific $T_3$ and $T_4$ antibodies. Plasma levels of hormones T3 and T4 were obtained as ng/mL.

## 4.3. Transcript Expression Quantification

Total pituitary RNA was extracted from tissues later conserved in RNA, using a total RNA purification kit (E.Z.N.A. Total RNA kit I, Omega bio-tek, Norcross, GA, USA) after mechanical tissue disruption. Total RNA was treated with DNase (DNA-free kit, Ambion (Thermo Fisher Scientific), Waltham, MA, USA) and its integrity and purity were assessed by 1% agarose gel electrophoresis. RNA was quantified using a NanoDrop 1000 Spectrophotometer (Thermo Fisher Scientific, Waltham, MA, USA). cDNA synthesis was carried out in a volume of 20 µl, containing 250 ng of DNase-treated RNA, 200 ng of random hexamers (Jena Biosciences, Thuringia, Germany), 100 U of RevertAid Reverse Transcriptase (Fermentas, Thermo Fisher Scientific, Waltham, MA, USA) and 8 U of RiboLock RNase Inhibitor (Fermentas). Reactions were incubated for 10 min at 25 °C and then 60 min at 42 °C, followed by enzyme inactivation for 10 min at 70 °C. cDNA was then stored at −20 °C until use. Primers used to amplify the pituitary hormones (*tsh*, *gh*, *pomca* and *pomcb*) and pituitary receptors (*trα*, *trβ* and *gr*) have been previously described in [56,57] Transcripts' abundance was measured by quantitative real-time PCR (qPCR) in individual pituitary cDNAs from each experimental group, using the relative standard curve method and EvaGreen chemistry. The copy numbers of target and reference genes was calculated as described in [58] using the equation:

$$\text{number of copies} = (X/NA)/(Y \times 1 \times 109 \times 650)$$

were X is the initial template amount (ng of the amplicon fragment), NA is the Avogadro's number, Y is the template length (bp of each amplicon), and 650 (Da) is the average weight of a base pair. Standard curves were prepared for each gene from serial dilutions of quantified amplicons and its efficiency was

always ≥90% and $R^2 \geq 0.99$. Each qPCR reaction contained 1x Sso Fast EvaGreen Supermix (Bio-Rad), 300 nM of each specific primer (Table 2) and 2 μL of cDNA (diluted 1:5 for target genes and 1:2500 for reference gene), in a final volume of 15 μL. A technical duplicate was performed for each cDNA sample. Two negatives for reverse transcriptase (-RT) were performed too. Reactions were run on a StepOnePlus qPCR thermocycler (Applied Biosystems (Thermo Fisher Scientific), Waltham, MA, USA) for 40 cycles, using the cycling conditions recommended by the supplier and the optimized annealing temperatures for each set of primers (Table 2). Specificity of qPCR reactions was confirmed by the presence of single peak in melt curve, amplification of expected size bands analysed in a 2% agarose gel electrophoresis and finally by amplicon sequencing. No amplification occurred in RT negatives, confirming the absence of genomic DNA contamination. All expression data was normalized to the housekeeping gene *18S.* Due to the low amount of cDNA available, only this gene was evaluated and its stability as a reference gene was confirmed in BestKeeper software [59]. Relative expression levels were calculated for each individual as the fold change compared to the pre-cold period sampling of low-lipid diet (D14).

**Table 2.** List of the amplified genes and the primers used for analysis of gene expression by quantitative qPCR.

| Pituitary Markers | Gene Name and Abbreviation | Fw/Rv [a] | Primer Sequence (5'-3') | Ta [b] | Bp [c] |
|---|---|---|---|---|---|
| Hormones | Growth Hormone (gh) | Fw | CCGAGGAACCAGATTTCACCCAA | 62 | 110 |
| | | Rv | TGGAGGGCGGAGCTATCAGGGA | | |
| | Thyroid stimulating hormone (tsh) | Fw | GTGTTCCCTTTCTGGCTCTTTTTTC | 52 | 100 |
| | | Rv | ACTCACACTCTGGTCTCTCCACGTA | | |
| | Proopiomelanocortin A (pomca) | Fw | CTTGAAGAAACCAAATGAACATC | 60 | 162 |
| | | Rv | GAAACAGCCAATGAAGACCTAA | | |
| | Proopiomelanocortin B (pomcb) | Fw | GCTCGTTAGCAGACCAAT | 62 | 76 |
| | | Rv | CAAAACACTCTCTCTTCATCTCT | | |
| Receptors | Thyroid receptor alfa (trα) | Fw | GAGGCCGGAGCCAAACAC | 60 | 124 |
| | | Rv | GCCGATATCATCCGACAGG | | |
| | Thyroid receptor beta (trβ) | Fw | ACCGACTGGAGCCCACACAG | 60 | 129 |
| | | Rv | CCTTCACCCACGCTGCACT | | |
| | Glucocorticoid receptor (gr) | Rv | CCATCACCTCTGCCGCATCTG | 64 | 195 |
| | | Fw | TCTGGAGGAACTGCTGCTGAACC | | |
| Reference | 18S ribosomal RNA (18s) | Fw | TGACGGAAGGGCACCACCAG | 60 | 158 |
| | | Rv | AATCGCTCCACCAACTAAGAACGG | | |

[a] Forward (Fw) or reverse (Rv) primers. [b] Optimized annealing temperature used for each pair of primers. [c] Amplicon size in base pairs (bp).

### 4.4. Statistical Analysis

Statistical differences between periods for each individual diet were analysed by nested one-way analysis of variance (ANOVA) with tank as a random factor to test for a possible tank effect. When a tank effect was not found, ANOVA followed by Tukey's or Dunnett's post-hoc test was used when variances were uniform or not, respectively. To compare diets within each period, Student's t-test was performed. Statistical differences were considered significant when p-values were less than 0.05. The Shapiro–Wilk test was previously used to ensure the normal distribution of data, and the uniformity of variances was determined by the statistical Levene test. Additionally, Principal component analysis (PCA) was performed, including $T_3$ and $T_4$ values together with growth parameters and plasma metabolites [8,16], respectively. All statistical analysis was performed using commercial software (PASW 20.0, SPSS Inc., Chicago, IL, USA).

## 5. Conclusions

This is the first study of pituitary response to thermal fluctuation in fish. Here, we demonstrate the capacity of the pituitary to recognize external inputs, such as dietary energy and water temperature, mediated by the hypothalamus, and to modulate its response via the expression of key hormones and

receptors. We have analysed the gene expression of hormones and hormone receptors as markers of pituitary activities and related them to overall fish response. At warm temperatures, whereas pituitary activity depends on dietary lipids/energy, fish can compensate for a 4% lipid reduction and maintain growth. In response to cold, the pituitary over-expressed *tsh* and *gh*, as well as some receptors, indicating resistance to global cold depression and capacity to integrate both external and internal inputs. In spite of the pituitary response at the transcription level, the sustained low temperatures appear to block the pituitary signals, and growth and metabolism were arrested. The mechanism underlying the gap between pituitary and whole-body responses is still unknown, and further studies are needed to elucidate the lack of linkage of the response at the pituitary hormone release level, to the target tissue levels or cold-depressed cell response level. All the cold effects are transitory and reverted when temperatures and ingesta are restored. Nonetheless, and in agreement with metabolism and redox data, animals fed less lipid (D14) experienced a higher stress condition, which delays or makes their recovery from low temperatures more difficult.

**Author Contributions:** S.S.-N. and S.C.S. performed the experiments, the data curation and writing of the original draft as well as the review and edition. B.O.-G., I.S. and L.F.-A. contributed to the data curation and to the writing-review and edition. A.I. and P.M.G. secured the resources, supervised the experiments, designed the trial and contribute to the writing of the original and final drafts. All authors contributed substantially to the work, agreed to be accountable for the content of the work, and agreed to be listed and approved the submitted version of the manuscript.

**Funding:** This study was supported by a grant from the Spanish government (AGL2011-29873), and by the FTC foundation grants from the Portuguese government (UID/Multi/04326/2013 and PTDC/BIA-ANM/4225/2012).

**Acknowledgments:** Authors thank Patricia Pinto for her guidelines on pituitary expression results interpretation and Ana Patricia Mateus for allowing the use of the primers of each analyzed gene.

**Conflicts of Interest:** The authors declare no conflict of interest.

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
