# Peer review of "Modulation of Pituitary Response by Dietary Lipids and Throughout a Temperature Fluctuation Challenge in Gilthead Sea Bream"

_fishes, doi:10.3390/fishes4040055_

Round 1

Reviewer 1 Report

Manuscript raises issues interesting for scientific community. Paper is well written, results are clearly described and discussed with relevant researches of other authors. 

Author Response

Manuscript raises issues interesting for scientific community. Paper is well written, results are clearly described and discussed with relevant researches of other authors.

Response: We thank the interest of the reviewer in our manuscript and we also thank his positive revision.

Reviewer 2 Report

The manuscript deal with the response of some sea bream pituitary hormones, measured by their gene expression, to different temperature and two diet lipid level. The topic is of interest for sea bream farmer and for sea bream feed producer, as they could suggest different diet with different energy level according to water temperature. I have some suggestions that could increase the quality of the article and authors have to correct some error before the publication of the article in fishes. 

The suggestion is to provide data about the feed intake of seabream during the thermal period. Fish eated less feed during cold season? Could you link your results also with this aspect?

Beside this suggestion here the errors that I found in the manuscript:

In figure 2 you have to change the order of histograms as in the text (line 128 you refer to figure 2A with gr but in the figure the histogram A is about trα.

Line 407 you named a fig 1 to explain the termal trial but Figure 1 presented is about gene expression linked to different lipid composition.

Author Response

The manuscript deal with the response of some sea bream pituitary hormones, measured by their gene expression, to different temperature and two diet lipid level. The topic is of interest for sea bream farmer and for sea bream feed producer, as they could suggest different diet with different energy level according to water temperature. I have some suggestions that could increase the quality of the article and authors have to correct some error before the publication of the article in fishes.

The suggestion is to provide data about the feed intake of seabream during the thermal period. Fish eated less feed during cold season? Could you link your results also with this aspect?

According with the reviewer suggestions, we have added data form SGR and FCR in lines 227-228, 314-316 and 366-369 of the tracked manuscript in each discussion sections. All data shown are published in Sanchez-Nuño et al. 2018 (Aquacult Environ Interact).

Beside this suggestion here the errors that I found in the manuscript:

In figure 2 you have to change the order of histograms as in the text (line 128 you refer to figure 2A with gr but in the figure the histogram A is about trα.

In accordance with the comment from the reviewer, the correction is applied in the tracked manuscript.

Line 407 you named a fig 1 to explain the termal trial but Figure 1 presented is about gene expression linked to different lipid composition.

In accordance with the comment from the reviewer, the correction is applied in the tracked manuscript.

Reviewer 3 Report

The paper gives an interesting information on pituitary response of gilthead sea bream juveniles (145 g initial weight) subjected to a simulated natural temperature and fed two dietary lipid levels. Experimental design is adequate to the aims of the performed experiment and results are properly discussed so I feel that is worth to be publised. However, I have some suggestions and questions that should be considered.

The last paragraph of Introduction (lines 108 to 110) should be moved to results. also in the first paragraph of discussion (lines 209 to 217) nothing is discussed, from my point of view deals more with material and methods section. 

I have some doubts about material and methods. Authors stated that they used fifteen 500-L tanks. If animals were distributed in two groups, each in triplicate, I understand that sea bream juveniles were maintained in six tanks. For what the other 9 tanks were used?

Another question, body weight and length of animal were measured but there is not information about in results section. Are there diffferences in growth due to different diets used?, if so, It could affect the pituitary response?

Also, although real feed intake was measured authors does not provide the obtained. Because one of the main effects of cold periods is the reduction of food intake,  data and discussion are important.

Author Response

The paper gives an interesting information on pituitary response of gilthead sea bream juveniles (145 g initial weight) subjected to a simulated natural temperature and fed two dietary lipid levels. Experimental design is adequate to the aims of the performed experiment and results are properly discussed so I feel that is worth to be publised. However, I have some suggestions and questions that should be considered.

The last paragraph of Introduction (lines 108 to 110) should be moved to results. also in the first paragraph of discussion (lines 209 to 217) nothing is discussed, from my point of view deals more with material and methods section.

According with the reviewer suggestion, we removed the last sentence of the introduction (lines 108-110). In addition, the first paragraph of discussion is moved to material and methods section (lines 400-408 of the tracked manuscript).

I have some doubts about material and methods. Authors stated that they used fifteen 500-L tanks. If animals were distributed in two groups, each in triplicate, I understand that sea bream juveniles were maintained in six tanks. For what the other 9 tanks were used?

The reviewer is right. We regret the mistake and the number of tanks used have been corrected to six.

Another question, body weight and length of animal were measured but there is not information about in results section. Are there diffferences in growth due to different diets used?, if so, It could affect the pituitary response?

Also, although real feed intake was measured authors does not provide the obtained. Because one of the main effects of cold periods is the reduction of food intake,  data and discussion are important.

According with the reviewer suggestions, we have added data form SGR and FCR in lines 227-228, 314-316 and 366-369 of the tracked manuscript in each discussion sections. All data shown are published in Sanchez-Nuño et al. 2018 (Aquacult Environ Interact).